# Learning Relation Representations from Word Representations

**Huda Hakami**

*Department of Computer Science, The University of Liverpool,*     H.A.HAKAMI@LIVERPOOL.AC.UK
*Liverpool, L69 3BX, UK*
*Department of Computer Science, Taif University,*                 HODA.H@TU.EDU.SA
*Saudi Arabia, Taif*

**Danushka Bollegala**                       DANUSHKA.BOLLEGALA@LIVERPOOL.AC.UK
*Department of Computer Science, The University of Liverpool,*
*Liverpool, L69 3BX, UK*

## Abstract

Identifying the relations that connect words is an important step towards understanding human languages and is useful for various NLP tasks such as knowledge base completion and analogical reasoning. Simple unsupervised operators such as vector offset between two-word embeddings have shown to recover some specific relationships between those words, if any. Despite this, how to accurately learn generic relation representations from word representations remains unclear. We model relation representation as a supervised learning problem and learn parametrised operators that map pre-trained word embeddings to relation representations. We propose a method for learning relation representations using a feed-forward neural network that performs relation prediction. Our evaluations on two benchmark datasets reveal that the penultimate layer of the trained neural network-based relational predictor acts as a good representation for the relations between words.

## 1. Introduction

Different types of relations exist between words in a language such as Hypernym, Meronym, Synonym, etc. Representing relations between words is important for various NLP tasks such as questions answering [Yang et al., 2017], knowledge base completion [Socher et al., 2013] and relational information retrieval [Duc et al., 2010].

Two main approaches have been proposed in the literature to represent relations between words. In the first approach, a pair of words is represented by a vector derived from a statistical analysis of a text corpus [Turney and Littman, 2005]. In a text corpus, a relationship between two words $\mathbf{X}$ and $\mathbf{Y}$ can be expressed using lexical patterns containing $\mathbf{X}$ and $\mathbf{Y}$ as slot variables. For example, "$\mathbf{X}$ is a $\mathbf{Y}$" or "$\mathbf{Y}$ such as $\mathbf{X}$" indicate that $\mathbf{Y}$ is a Hypernym of $\mathbf{X}$ [Snow et al., 2005]. The elements of the vector representing the relation between two words correspond to the number of times those two words co-occur with a particular pattern in a corpus. Given such a relation representation, the relational similarity between the relations that exist between the two words in two word-pairs can be measured by the cosine of the angle between the corresponding vectors. We call this the *holistic* approach because a pair of words is treated as a whole rather than the two constituent words separately when creating a relation representation [Turney, 2005]. Sparsity is a well-

known problem for the holistic approach as two words have to co-occur enough in a corpus, or else no relation can be represented for rare or unseen word-pairs.

In contrast, the second approach for relation representation directly computes a relation representation from pre-trained word representations (i.e. word embeddings) using some *relational operators.* Prediction-based word embedding learning methods [Pennington et al., 2014, Mikolov et al., 2013a] represent the meaning of individual words by dense, low-dimensional real-valued vectors by optimising different language modelling objectives. Although no explicit information is provided to the word embedding learning algorithms regarding the semantic relations that exist among words, prior work [Mikolov et al., 2013b] has shown that the learnt word embeddings encode remarkable structural properties pertaining to semantic relations. They showed that the difference (vector offset) between two word vectors (here-onwards denoted by PAIRDIFF) is an accurate method for solving analogical questions in the form "$a$ is to $b$ as $c$ is to ?". For example, $\boldsymbol{king} - \boldsymbol{man} + \boldsymbol{woman}$ results in a vector that is closest to the $\boldsymbol{queen}$ vector. We call this approach *compositional* because the way in which the relation representation is composed by applying some linear algebraic relational operator on the the semantic representations of the the words that participate in a relation. This interesting property of word embeddings sparked a renewed interest in methods that compose relation representations using word embeddings and besides PairDiff, several other unsupervised methods have been proposed such as 3CosAdd and 3CosMult [Levy and Goldberg, 2014].

Despite the initial hype, recently, multiple independent works have raised concerns about of word embeddings capturing relational structural properties [Linzen, 2016, Schluter, 2018, Liu et al., 2018, Rogers et al., 2017, Gladkova et al., 2016]. Although PairDiff performs well on the Google analogy dataset, its performance for other relation types has been poor [Chen et al., 2017, Vylomova et al., 2016, Köper et al., 2015]. Vylomova et al. [2016] tested for the generalisation ability of PAIRDIFF using different relation types and found that semantic relations are captured less accurately compared to syntactic relations. Likewise, Köper et al. [2015] showed that word embeddings are unable to detect paradigmatic relations such as Hypernym, Synonym and Antonyms. Methods such as PAIRDIFF are biased towards attributional similarities between individual words than relational similarities and fails in the presence of nearest neighbours. We further discuss various limitations of the existing unsupervised relation composition methods in Section 2.2.

Considering the above-mentioned limitations of the unsupervised relation composition methods, a natural question that arises is whether it is possible to learn *supervised* relation composition methods to overcome those limitations. In this paper, we model relation representation as learning a parametrised operator $f(\boldsymbol{a}, \boldsymbol{b}; \theta)$ such that we can accurately represent the relation between two given words $a$ and $b$ from their word representations $\boldsymbol{a}$ and $\boldsymbol{b}$, without modifying the input word embeddings. For this purpose, we propose a Multi-class Neural Network Penultimate Layer (**MnnPl**), a simple and effective parametrised operator for computing relation representations from word representations. Specifically, we train a nonlinear multilayer feed-forward neural network using a labelled dataset consisting of word-pairs for different relation types, where the task is to predict the relation between two input words represented by their pre-trained word embeddings. We find that the penultimate layer of the trained neural network provides an accurate relation representation that generalises beyond the relations in the training dataset. We emphasise that our focus here

is not to classify a given pair to a relation in a pre-defined set (relation classification), but rather to obtain a good representation for the relation between the two words in the pair. Our experimental results show that **MnnPl** significantly outperforms unsupervised relational operators including PairDiff in two standard benchmark datasets, and generalises well to unseen out-of-domain relations.

## 2. Related Work

Relations between words can be classified into two types namely, *contextual* and *lexical* [Hendrickx et al., 2009, Nastase et al., 2013, Gábor et al., 2017]. Contextual relations are relations that exist between two words in a given specific context such as a sentence. For example, given the sentence "the *machine* makes a lot of *noise*", a Cause-Effect relation exists between the *machine* and *noise* in this particular sentence. More examples of *Contextual* relations can be found in Hendrickx et al. [2009]. On the other hand, *Lexical* relations hold between two words independent of the contexts in which those two words occur. For instance, the lexical relation capital-of exists between *London* and *England*. WordNet, for example, organises words into various lexical relations such as is-a-synonym-of, is-a-hypernym-of, is-an-meronym-of, etc. Our focus in this paper is on representing *lexical* relations.

### 2.1 Relation Representation Operators

Word embeddings learning methods map words to real-valued vectors that represent the meanings of those words. Given the embeddings of two words, Mikolov et al. [2013b] showed that relations that hold between those words can be represented by the vector-offset (difference) between the corresponding word embeddings. This observation sparked a line of research on relational operators that can be used to discover relational information from word embeddings besides vector-offset. Using pre-trained word embeddings to represent relations is attractive for computational reasons. Unlike holistic approaches that represent the relation between two words by lexico-syntactic patterns extracted from the co-occurrence contexts of the two words, relational operators do not require any co-occurrence contexts. This is particularly attractive from a computational point of view because the number of possible pairings of $n$ words grows $\mathcal{O}(n^2)$, implying that we must retrieve co-occurrence contexts for all such pairings for extracting lexico-syntactic patterns for the purpose of representing the relations between words. On the other hand, in the compositional approach, once we have pre-trained the word embeddings we can compute the relation representations for any two words without having to re-learn anything. For example, in applications such as relational search [Duc et al., 2011], we must represent the relation between two words contained in a user query. Because we cannot anticipate all user queries and cannot pre-compute relation representations user queries offline, relation compositional methods are attractive for relational search engines.

Compositional methods of relation representation differ from Knowledge Graph Embedding (KGE) methods such as TransE [Bordes et al., 2013], DistMult [Yang et al., 2015], CompIE [Trouillon et al., 2016], etc. in the sense that in KGE, given a knowledge graph of tuples $(h, r, t)$ in which a relation $r$ relates the (head) entity $h$ to the (tail) entity $t$, we must jointly learn embeddings for the entities as well as for the relations such that some

scoring function is optimised. For example, TransE scores a tuple $(h, t, r)$ by the $\ell_1$ or $\ell_2$ norm of the vector $(\boldsymbol{h} + \boldsymbol{r} - \boldsymbol{t})$ (we use bold fonts to denote vectors throughout the paper). On the other hand, the relation composition problem that we consider in this paper does not attempt to learn entity embeddings or relation embeddings from scratch but use pre-trained word/entity embeddings to compose relation representations. Therefore, compositional methods of relation representation are attractive from a computational point of view because we no longer need to learn the word/entity embeddings and can focus only on the relation representation learning problem.

On the other hand, compositional methods for relation representation differ from those proposed for solving the analogy completion such as 3CosAdd [Mikolov et al., 2013b], 3CosMult [Levy and Goldberg, 2014], 3CosAvg and LRCos [Drozd et al., 2016]. Analogy completion is the task of finding the missing word $(d)$ in the two analogical word-pairs "$a$ is to $b$ as $c$ is to $d$". To solve analogy completion, one must first detect the relation in which the two words in the first pair (i.e. $(a, b)$) stand in, and then find the word $d$ that is related in the same way to $c$. Methods that solve analogy questions typically consider the distances between the words of the two pairs in some common vector space. For example, 3CosAdd computes the inner product between the vector $(\boldsymbol{b} - \boldsymbol{a} + \boldsymbol{c})$ and the word embedding $\boldsymbol{d}$ for each word in the vocabulary. If the vectors are $l_2$ normalised then inner-product is equivalent to cosine similarity, which can be seen as a calculation involving cosine similarity scores for three pairs of words $(\boldsymbol{b}, \boldsymbol{d}), (\boldsymbol{a}, \boldsymbol{d})$ and $(\boldsymbol{c}, \boldsymbol{d})$ explaining its name 3CosAdd. 3CosMult, on the other hand, considers the same three cosine similarity scores but in a multiplicative formula. However, analogy completion methods such as 3CosAdd or 3CosMult cannot be considered as relation representation methods because they do not create a representation for the relation between $a$ and $b$ at any stage during the computation.

Hakami and Bollegala [2017a] compared different unsupervised relational operators such as PAIRDIFF, concatenation, vector addition and elementwise multiplication and reported that PAIRDIFF to be the best operator for analogy completion whereas, elementwise multiplication was the best for link prediction in knowledge graphs. A recent work [Hakami et al., 2018] has theoretically proven that PAIRDIFF to be the best linear unsupervised operator for relation representation when the relational distance (similarity) between two word-pairs is measured in term of the squared Euclidean distance between the corresponding relation representation vectors.

## 2.2 Limitations of Unsupervised Relational Operators

Recently, several limitations have been reported of the existing unsupervised relational representation operators [Linzen, 2016, Rogers et al., 2017]. In particular, the distance between word embeddings in a semantic space significantly affects the performance of PAIRDIFF in analogy completion. Specifically, to measure the relational similarity between $(a, b)$ and $(c, d)$ pairs using PAIRDIFF, prior work compute the inner-product between the normalised offset vectors: $(\boldsymbol{a} - \boldsymbol{b})^\top (\boldsymbol{c} - \boldsymbol{d})$. This is problematic because the task of measuring relational similarity between the two word-pairs is simply decomposed into a task of measuring lexical similarities between individual words of the pairs. Specifically, the above inner-product can be rewritten as $\boldsymbol{a}^\top \boldsymbol{c} - \boldsymbol{a}^\top \boldsymbol{d} - \boldsymbol{b}^\top \boldsymbol{c} + \boldsymbol{b}^\top \boldsymbol{d}$. This value can become large, for example when

$a$ is highly similar to $c$ or $b$ is highly similar to $d$, irrespective of the relationship between $a$ and $b$, and $c$ and $d$.

As a concrete example of this issue, consider measuring the relational similarity between (*water*, *riverbed*) and each of the two word-pairs (*traffic*, *street*) and (*water*, *drink*)). In this case, *water* flows-In *riverbed* is the implicit relation expressed by the two words in the stem word-pair (*water*, *riverbed*). Therefore, the candidate pair (*traffic*, *street*) is relationally more similar to the stem word-pair than (*water*, *drink*) because flows-In also holds between *traffic* and *street*. However, if we use pre-trained GloVe word embeddings [Pennington et al., 2014] with PAIRDIFF as the relation representation, then (*water*, *drink*) reports a higher relational similarity score (0.62) compared to that for (*traffic*, *street*) (0.42) because of the lexical similarities between the individual words.

PairDiff was originally evaluated by Mikolov et al. [2013b] using semantic and syntactic relations in the Google dataset such as Capital-City, Male-Female, Currency, City-in-State, singular-plural, etc. However, more recent works have shown that although PairDiff can accurately represent the relation types in the Google dataset, it fails on other types of relations [Köper et al., 2015, Fu et al., 2014]. For example, Köper et al. [2015] showed PAIRDIFF cannot detect paradigmatic relations such as hypernymy, synonymy and antonymy, whereas Fu et al. [2014] reported that hypernym-hyponym relation is more complicated and a single offset vector cannot completely represent it.

The space of unsupervised operators proposed so far in the literature is limited in the sense that the operators pre-defined and fixed, and cannot be adjusted to capture the actual relations that exist between words. It is unrealistic to assume that the same operator can represent all relation types from the word embeddings learnt from different word embedding learning algorithms. On the other hand, there are many datasets such as SemEval 2012 Task2, Google, MSR, SAT verbal analogy questions etc., which already provide examples of the types of relations that actually exist between words. Our proposed supervised relational composition method learns a parametrised operator implemented as a neural network, which can be trained to better represent relations between words.

### 2.3 Relation Detection using Word Embeddings

Word embeddings have been used as features in prior work for learning *lexical* relations between words. Given two words, Vylomova et al. [2016] first represent the relation between those words using PAIRDIFF and then train a multi-class classifier for classifying different relation types. Methods that focus on detecting a particular type of relation between two words such as hypernymy, have also used unsupervised relation composition operators such as PAIRDIFF to create a feature vector for a word-pair [Carmona and Riedel, 2017, Levy et al., 2015b, Roller et al., 2014, Fu et al., 2014]. Carmona and Riedel [2017] and Roller et al. [2014] train respectively a logistic regression classifier and a linear support vector classifier using word-pairs represented by the PAIRDIFF or concatenation of the corresponding pre-trained word embeddings.

Levy et al. [2015b] used PAIRDIFF and vector concatenation as operators for representing the relation between two words and evaluated the representations in a lexical entailment task and a hypernym prediction task. They found that the representations produced by these operators did *not* capture relational properties but simply retained the information

in individual words, which was then used by the classifiers to make the predictions. Similarly, Fu et al. [2014] observe that PAIRDIFF is inadequate to induce the hypernym relation. Chen et al. [2017] analysed PAIRDIFF on a number of different relation types and found that its performance varies significantly across relations.

## 3. Supervised Relational Operators

Our goal in this paper is to learn a parametrised two-argument function $f(\cdot, \cdot; \theta)$ that can accurately represent the relation between two given words $a$ and $b$ using their pre-trained $d$-dimensional word embeddings $\boldsymbol{a}, \boldsymbol{b} \in \mathbb{R}^d$. Here, $\theta$ denotes the set of parameters that governs the behaviour of $f$, which can be seen as a *supervised* operator that outputs a relation representation from two input word representations. The output of $f$, for example, could be a vector that exists in the same or a different vector space as $\boldsymbol{a}$ and $\boldsymbol{b}$, as given by (1).

$$f(\boldsymbol{a}, \boldsymbol{b}; \theta) : \mathbb{R}^d \times \mathbb{R}^d \to \mathbb{R}^m \tag{1}$$

In general $d \neq m$ and word and relation representations can have different dimensionalities and even when $d = m$ they might be in different vector spaces. We could extend this definition to include higher-order relation representations such as matrices or tensors but doing so would increase the computational overhead. Therefore, we limit supervised relational operators such that they return vectors as given by (1) in this paper. We note that unsupervised relational operators such as PAIRDIFF and vector concatenation are specific instances of this definition. For example, for PAIRDIFF we have $f(\boldsymbol{a}, \boldsymbol{b}; \theta) = \boldsymbol{a} - \boldsymbol{b}$ ($m = d$) , and for vector concatenation we have $f(\boldsymbol{a}, \boldsymbol{b}; \theta) = \boldsymbol{a} \oplus \boldsymbol{b}$ ($m = 2d$), where $\oplus$ denotes concatenation of two vectors. In unsupervised operators, $\theta$ is a constant that does not influence the output relation embedding.

### 3.1 Relation Representation via Relation Prediction

We implement the proposed supervised relation composition operator as a feed-forward neural network with one or more hidden layers followed by a softmax layer as shown in Figure 1. Weight matrices for the hidden layers are $\mathbf{W}_1$ and $\mathbf{W}_2$, whereas the biases are $\boldsymbol{s}_1$ and $\boldsymbol{s}_2$. $g$ refers to the nonlinear activation for the hidden layers. We experiment with different nonlinearities in the hidden layers. Using a dataset $\mathcal{D} = \{(a_i, r_i, b_i)\}_{i=1}^N$ of word-pairs $(a_i, b_i)$ with relations $r_i$, we train the neural network to predict $r_i$ given the concatenated pre-trained word embeddings $\boldsymbol{a}_i \oplus \boldsymbol{b}_i$ as the input. We minimise the $\ell_2$ regularised cross-entropy loss over the training instances. After training the neural network, we use its penultimate layer (i.e. the output of the final hidden layer) as the relation representation for a word-pair. We call this method Multi-class Neural Network Penultimate Layer (**MnnPl**).

We emphasise that our goal is *not* to classify a given pair into a specific set of relations, but rather to find a representation of the relation between any pair of words. Therefore, we test the learnt relation representation using relations that are not seen during training (i.e. out-of-domain examples) by holding out a subset of relations during training.

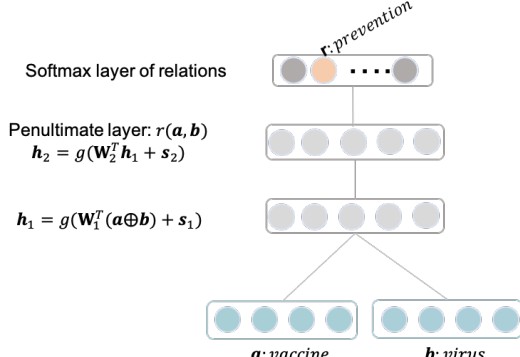

Figure 1: The framework of Multi-class Neural Network Penultimate Layer (**MnnPl**), a feed forward neural network that is used to model the supervised relational operator $f$.

## 4. Experiments and Results

We evaluate the relation embeddings learnt by the proposed **MnnPl** on two standard tasks: out-of-domain relation prediction and measuring the degree of relational similarities between two word-pairs. In Section 4.1, we first introduce the relational training datasets and the input word embedding models that we used to compose relation embeddings. Next, in Section 4.2, we describe the experimental setup that we follow to train the proposed method. We compare the performance of the **MnnPl** with various baseline methods as illustrated in Section 4.3. In Section 4.4 and 4.5, we discuss the experiments conducted on the out-of-domain and in-domain relation prediction task, respectively. The task of measuring the degree of relational similarities is presented in Section 4.6. In short, each two word-pairs in the dataset for this task has a manually assigned relational similarity score, which we consider as the gold standard rating for relational similarity.

### 4.1 Training Dataset and Word Embeddings

We used two previously proposed datasets for evaluating **MnnPl**: BATS[1] [Gladkova et al., 2016] and DiffVec[2] [Vylomova et al., 2016]. BATS is a balanced dataset that contains 4 main relation types, two are semantic relations (Lexicographic and Encyclopaedic) and the other two are syntactic relations (Inflectional and Derivational). Each main category has 10 different sub-relation types and 50 word-pairs are provided for each relation (2,000 unique word-pairs in total). DiffVec covers 36 subcategories that are classified into 15 main relation types in total (31 semantic and 6 syntactic). The dataset is unbalanced because a different number of word-pair examples assigned to each relation, in total it has 12,452 word-pairs. We exclude relations that has less than 10 examples from experiments.

For word embeddings, we use CBOW, Skip-Gram (SG) [Mikolov et al., 2013a] and GloVe [Pennington et al., 2014] as the input to the proposed method. For consistency of the comparison, we train all word embedding learning methods on the ukWaC corpus [Ferraresi et al., 2008], a web-derived corpus of English consisting of ca. 2 billion words. Words

---

1. http://vecto.space/projects/BATS/
2. https://github.com/ivri/DiffVec

that appear less than 6 times in the entire corpus are truncated, resulting in a vocabulary of 1,371,950 unique words. We use the publicly available implementations by the original authors for training the word embeddings using the recommended parameters settings. Specifically, GloVe model was trained with window size 15, 50 iterations, weighting function parameters $x_{\max} = 100, \alpha = 0.75$. CBOW and SG embeddings were trained with window size 8, 25 negative samples, 15 iterations, sampling parameter equal to $10^{-4}$.

In addition to the prediction-based word embeddings created using CBOW and GloVe, we use Latent Semantic Analysis (LSA) to obtain counting-based word embeddings [Deerwester et al., 1990, Turney and Pantel, 2010, Clark, 2015]. A co-occurrence matrix $\mathbf{M} \in \mathbb{R}^{n \times n}$ is first constructed considering the 50k most frequent words in the corpus to avoid data sparseness. The raw counts are weighted following positive point-wise mutual information (PPMI) method. Subsequently, singular value decomposition (SVD) is applied to reduce the dimensionality $\mathbf{M}$ to lower rank matrices $\mathbf{U}_k \mathbf{S}_k \mathbf{V}_k^\top$, where $\mathbf{S}_k$ is a diagonal matrix that has the largest $k$ singular values of $\mathbf{M}$ as the diagonal elements. $\mathbf{U}_k$ and $\mathbf{V}_k$ are orthogonal matrices of singular vectors of the corresponding $k$ singular values. Following Levy et al. [2015a], $\mathbf{S}_k$ is ignored when representing the words (i.e. $\mathbf{M} = \mathbf{U}_k$).

### 4.2 Training Details

We use the word embeddings trained on the ukWaC with 50 dimensions as the input to the neural network. Overall, we found $\ell_2$ normalisation of word embeddings to improve results. We use Stochastic Gradient Descent (SGD) with Momentum [Qian, 1999] with mini-batch size of 128 to minimise the $\ell_2$ regularised cross-entropy error. All parameters are initialised by uniformly sampling from $[-1, +1]$ and the initial learning rate is set to 0.1. Dropout regularisation is applied with a 0.25 rate. Tensorflow is used to implement the model. We train the models till the convergence on a validation split. We used the Scholastic Aptitude Test (SAT) 374 multiple choice analogy questions dataset [Turney et al., 2003] for validating the hyperparameter values. Specifically, we selected the number of the hidden layers among $\{1, 2, 3\}$ and the activation function $g$ of the hidden layers among {tanh, relu, linear}. On the validation dataset, we found the optimal configuration was to set the number of hidden layers to two and the nonlinear activation to tanh. The optimal $\ell_2$ regularisation coefficient $\lambda$ was 0.001. We train the models till the convergence on the validation dataset. These settings performed consistently well in all our evaluations.

### 4.3 Baseline Methods

We compare the relation representations produced by **MnnPl** against several baselines as detailed next. Note that the considered baselines produce relation representations for word-pairs.

**Unsupervised Baselines:** We implement the following unsupervised relational operators for creating relation representations using word embeddings Hakami and Bollegala [2017b]: PAIRDIFF, Concatenation (CONCAT), elementwise addition (ADD) and elementwise multiplication (MULT). These operators are unsupervised in the sense that there are no parameters in those operators that can be learnt from the training data.

**Supervised Baselines:**  We design a supervised version of the CONCAT operator parametrised by a weight matrix $\mathbf{W} \in \mathbb{R}^{d \times m}$ and a bias vector $\boldsymbol{s} \in \mathbb{R}^m$ to compute a relation representation $\boldsymbol{r}$ for two words $\boldsymbol{a}$ and $\boldsymbol{b}$ as given in (2).

$$r(\boldsymbol{a}, \boldsymbol{b}; (\mathbf{W}, \boldsymbol{s})) = \mathbf{W}^\top (\boldsymbol{a} \oplus \boldsymbol{b}) + \boldsymbol{s} \tag{2}$$

We call this baseline as the Supervised Concatenation (**Super-Concat**).

Likewise, we design a supervised version of PAIRDIFF, which we name **Super-Diff** as follows:

$$r(\boldsymbol{a}, \boldsymbol{b}) = \mathbf{W}^\top (\boldsymbol{b} - \boldsymbol{a}) + \boldsymbol{s} \tag{3}$$

In addition to the above supervised operators, we use the bilinear operator proposed by [Hakami et al., 2018] (given in (4)) as a supervised relation representation method.

$$r(\boldsymbol{a}, \boldsymbol{b}) = \boldsymbol{a}^\top \underline{\mathbf{A}} \boldsymbol{b} + \mathbf{P} \boldsymbol{a} + \mathbf{Q} \boldsymbol{b} + \boldsymbol{s} \tag{4}$$

Here, $\underline{\mathbf{A}} \in \mathbb{R}^{d \times d \times m}$ is a 3-way tensor in which each slice is a $d \times d$ real matrix. The first term in (4) corresponds to the pairwise interactions between $\boldsymbol{a}$ and $\boldsymbol{b}$. $\mathbf{P}, \mathbf{Q} \in \mathbb{R}^{d \times d}$ are the projection matrices involving first-order contributions respectively of $\boldsymbol{a}$ and $\boldsymbol{b}$ towards $r$. We refer to this operator as **BiLin**.

We train the above-mentioned three supervised relational operators using a margin-based rank loss objective. Specifically, we minimise the distance between the relation representations of the analogous pairs (positive instances), while maximising the distance between the representations of non-analogous examples (negative instances) created via random perturbations. Given a set of word pairs $\mathcal{S}_r$ that are related by the same relation, we generate positive training instances $((a, b), (c, d))$ by pairing word-pairs $(a, b) \in \mathcal{S}_r$ and $(c, d) \in \mathcal{S}_r$. Next, to generate negative training instances, we corrupt a positive instance by pairing $(a, b) \in \mathcal{S}_r$ with a word-pair $(c', d') \in \mathcal{S}_{r'}$ that belongs to a different relation $r' \neq r$. One negative instance is generated for each analogous example in our experiments, resulting in a balanced binary labelled dataset. The regularised training objective $\mathcal{L}(\mathcal{D}; \theta)$ is given by (5).

$$\sum_{((a,b),(c,d),(c',d')) \in \mathcal{D}} \max(0, \delta + ||r(\boldsymbol{a}, \boldsymbol{b}) - r(\boldsymbol{c}, \boldsymbol{d})||_2^2 - ||r(\boldsymbol{a}, \boldsymbol{b}) - r(\boldsymbol{c'}, \boldsymbol{d'})||_2^2) + \frac{\lambda}{2} ||\theta||_2^2 \tag{5}$$

Here, $\delta$ is a margin hyperparameter set to 1 according to the best accuracy on the SAT validation dataset.

The regularisation coefficient $\lambda$ is set separately for each parameter in the different supervised relation composition operators using the SAT validation dataset. For **Super-Concat** and **Super-Diff**, regularising $\mathbf{W}$ and $\mathbf{s}$ resulted in lowering the accuracy on SAT questions. Therefore, no regularisation is applied for those two operators. For the **BLin** operator, the best regularisation coefficient for the tensor $\underline{\mathbf{A}}$ on the validation dataset was 0.1. However, regularising $\mathbf{P}$ and $\mathbf{Q}$ decreased the performance on the validation set, and therefore were not regularised.

### 4.4 Out-of-domain Relation Prediction

A critical evaluation criterion for a relation representation learning method is whether it can accurately represent not only the relations that exist in the training data that was used to learn the relation representation but can also generalise to unseen relations (*out-of-domain*). Therefore, to evaluate the different relation representation methods, we employ them in an out-of-domain relation prediction task. Specifically, we use different relations for testing than that used in training. No training is required for unsupervised operators.

Next, we describe the evaluation protocol in detail. Lets denote a set of relation types by $\mathcal{R}$ and a set of word-pairs covering the relations in $\mathcal{R}$ by $\mathcal{D}$. First, we randomly sample five target relations from the dataset to construct a relation set $\mathcal{R}_t$ for testing and the remainder represents a set of source relations $\mathcal{R}_s$ that is used for training the supervised relational operators including the supervised baselines and the proposed **MnnPl**. We use the set $\mathcal{D}_s$ of word-pair instances covering $\mathcal{R}_s$ to learn the supervised operators by predicting the relations in $\mathcal{R}_s$. To evaluate the performance of such operators, we use the relational instances in the test split $\mathcal{D}_t$ that cover the out-of-domain relations in $\mathcal{R}_t$. We conduct 1-NN relation classification on $\mathcal{D}_t$ dataset. The task is to predict the relation that exists between two words $a$ and $b$ from the sampled relations in $\mathcal{R}_t$. Specifically, we represent the relation between two words using each relational operator on the corresponding word embeddings. Next, we measure the cosine similarity between representations for the stem pair and all the word-pairs in $\mathcal{D}_t$. For each target word-pair, if the top-ranked word-pair has the same relation as the stem pair, then it is considered to be a correct match. Note that we do *not* use $\mathcal{D}_t$ for learning or updating the (supervised) relational operator but use it only for the 1-NN relation predictor. We repeat this process ten times by selecting different $\mathcal{R}_s$ and $\mathcal{R}_t$ relation sets and use leave-one-out evaluation for the 1-NN as the evaluation criteria. We compute the (micro-averaged) classification accuracy of the test sets as the evaluation measure. Because each relation type in an out-of-domain relation set has multiple relational instances, a suitable relation representation method retrieves the related pairs for a target pair at the top of the ranked list. For this purpose, we measure Mean Average Precision (MAP) for the relation representation methods.

To derive further insights into the relation representations learnt, following Nastase et al. [2013], we use the notion of "near" vs. "far" analogies considering the similarities between the corresponding words in the two related pairs. For example, (*tiger*, *feline*), (*cat*, *animal*) and (*motorcycle*, *vehicle*) are all instances of the IS-A-HYPERNYM-OF relation. One could see that (*tiger*, *feline*) is closer to (*cat*, *animal*) than (*motorcycle*, *vehicle*). Here, *tiger* and *cat* are similar because they are both animals; also *feline* and *animal* have shared attributes. On the other hand, the corresponding words in the two pairs (*tiger*, *feline*) and (*motorcycle*, *vehicle*) have low attributional similarities between *tiger* and *motorcycle* or between *feline* and *vehicle*. Detecting near analogies using word embeddings is easier compared to far analogies because attributional similarity can be measured accurately using word embeddings. For this reason, we evaluate the accuracy of a relation representation method at different degrees of the analogy as follows. Given two word-pairs, we compute the cross-pair attributional similarity using SimScore defined by (6).

$$\text{SimScore}((a, b), (c, d)) = \frac{1}{2}(\text{sim}(\boldsymbol{a}, \boldsymbol{c}) + \text{sim}(\boldsymbol{b}, \boldsymbol{d})) \tag{6}$$

Here, $\text{sim}(\boldsymbol{x}, \boldsymbol{y})$ is the cosine similarity between $\boldsymbol{x}$ and $\boldsymbol{y}$. Next, we sort the word-pairs in the descending order of their SimScores (i.e. from near to far analogies). Examples of far and near analogies with SimScores for some selected word-pairs are presented in Table 1. To alleviate the effect of attributional similarity between two word-pairs in our evaluation, we remove the 25% top-ranked (nearest) pairs for each stem pair. Consequently, a relation representation method that relying only on attributional similarity is unlikely to accurately represent the relations between words.

| Relation type | Stem pair | Nearest to Farthest |
|---|---|---|
| Hyper | (food:cherry) | (fruit:plum)$_{0.87}$,(veggie:parsley)$_{0.81}$, . . . , (gun:cannon)$_{0.56}$,(artifact:helicopter)$_{0.43}$ |
| Space-Time | (theatre:play) | (hall:music)$_{0.69}$,(studio:art)$_{0.68}$, . . . , (diary:milk)$_{0.42}$,(mine:coal)$_{0.38}$ |
| Cause-Effect | (disease:sickness) | (illness:discomfort)$_{0.84}$,(headache:stress)$_{0.78}$, . . . , (question:answer)$_{0.51}$,(digging:hole)$_{0.47}$ |
| Contiguity | (wall:shelf) | (sill:window)$_{0.78}$,(railing:stair)$_{0.76}$, . . . , (mountain:valley)$_{0.59}$,(margin:paper)$_{0.59}$ |

Table 1: The two nearest and the two farthest word-pairs for some stem word-pairs along with their similarity scores according to Equation 6.

| | CBOW | | | | SG | | | |
|---|---|---|---|---|---|---|---|---|
| | DiffVec | | BATS | | DiffVec | | BATS | |
| Method | Acc | MAP | Acc | MAP | Acc | MAP | Acc | MAP |
| Super-Concat | 0.371 | 0.354 | 0.712 | 0.586 | 0.356 | 0.309 | 0.595 | 0.486 |
| Super-Diff | 0.361 | 0.288 | 0.565 | 0.425 | 0.321 | 0.283 | 0.486 | 0.362 |
| BiLin | 0.364 | 0.329 | 0.710 | 0.587 | 0.332 | 0.309 | 0.604 | 0.485 |
| PairDiff | 0.397 | 0.344 | 0.688 | 0.525 | 0.349 | 0.305 | 0.607 | 0.454 |
| Concat | 0.173 | 0.347 | 0.325 | 0.518 | 0.147 | 0.316 | 0.250 | 0.446 |
| Add | 0.164 | 0.302 | 0.321 | 0.479 | 0.159 | 0.288 | 0.269 | 0.412 |
| Mult | 0.179 | 0.213 | 0.330 | 0.286 | 0.206 | 0.24 | 0.289 | 0.287 |
| **MnnPl** | **0.486** | **0.421** | **0.721** | **0.624** | **0.411** | **0.373** | **0.625** | **0.522** |
| | GloVe | | | | LSA | | | |
| | DiffVec | | BATS | | DiffVec | | BATS | |
| Method | Acc | MAP | Acc | MAP | Acc | MAP | Acc | MAP |
| Super-Concat | 0.338 | 0.329 | 0.673 | 0.553 | 0.274 | 0.265 | 0.649 | 0.56 |
| Super-Diff | 0.313 | 0.275 | 0.540 | 0.41 | 0.282 | 0.253 | 0.536 | 0.424 |
| BiLin | 0.355 | 0.325 | 0.668 | 0.557 | 0.263 | 0.269 | 0.622 | 0.543 |
| PairDiff | 0.365 | 0.312 | 0.663 | 0.516 | 0.295 | 0.306 | 0.624 | 0.51 |
| Concat | 0.139 | 0.3 | 0.361 | 0.52 | 0.122 | 0.3 | 0.298 | 0.482 |
| Add | 0.161 | 0.276 | 0.347 | 0.462 | 0.132 | 0.266 | 0.312 | 0.442 |
| Mult | 0.199 | 0.225 | 0.323 | 0.278 | 0.179 | 0.198 | 0.385 | 0.335 |
| **MnnPl** | **0.456** | **0.381** | **0.698** | **0.585** | **0.360** | **0.342** | **0.658** | **0.59** |

Table 2: Average accuracy of 1-NN relation classification for different relation representation methods on DiffVec and BATS datasets. Results are shown for CBOW, SG, GloVe and LSA word embeddings (50 dimensional embeddings).

|              | Encyclopedic | | Lexicographic | |
| --- | --- | --- | --- | --- |
| Method       | Acc   | MAP   | Acc   | MAP   |
| Super-Concat | 0.829 | 0.72  | 0.294 | 0.268 |
| Super-Diff   | 0.573 | 0.446 | 0.296 | 0.226 |
| BiLin        | 0.813 | 0.694 | 0.366 | 0.3   |
| PairDiff     | 0.764 | 0.613 | 0.297 | 0.213 |
| Concat       | 0.724 | 0.792 | 0.146 | 0.302 |
| Add          | 0.774 | 0.781 | 0.199 | 0.311 |
| Mult         | 0.464 | 0.294 | 0.170 | 0.202 |
| **MnnPl**    | **0.884** | **0.813** | **0.414** | **0.338** |

Table 3: Break down the performance for the two semantic relation types in BATS per method using GloVe embeddings.

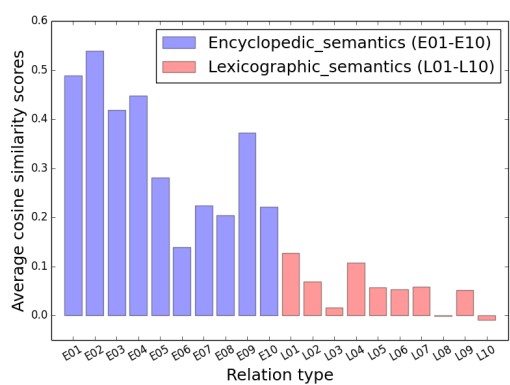

Figure 2: Average cosine similarity between PairDiff embeddings for different relation types in BATS.

The average accuracy (Acc) and the MAP of the relation representation operators for CBOW, SG, GloVe and LSA embeddings are presented in Table 2. As can be observed among the different embedding types, **MnnPl** consistently outperforms all other methods in both Acc and MAP score. The differences between **MnnPl** and other methods for all rounds and target relations are statistically significant ($p < 0.01$) according to a paired t-tes. CBOW embeddings report the best Acc and MAP scores for the two datasets in contrast to all other embedding models. We also assess how good such relational operators are on the in-domain relation prediction task, wherein the task is to represent relational instances that belong to the relation set used on training the models. We find that **MnnPl** can accurately represent relations in this in-domain setting as well (see Section 4.5).

To further evaluate the accuracy of the different relational operators on different relation types, we break down the evaluation per major semantic relation type in the BATS dataset as shown in Table 3. We see that lexicographic relations are more difficult compared to encyclopaediac relations for all methods. Overall, the proposed **MnnPl** consistently outperforms other methods for both types of semantic relations. On the other hand, PairDiff performs significantly worse for lexicographic relations. We believe that this result explains PairDiff's superior performance on the Google analogy dataset, which contains a large proportion of encyclopaediac relations such as capital-common-countries, capital-currency, city-in-state, and family. ADD achieves the second best accuracy for Encyclopedic relations (where PairDiff is only slightly behind it), whereas Concat follows **MnnPl** in term of MAP scores. For encyclopaediac relations, the head words can be grouped into a sub-space in the embedding space that is roughly aligned with the sub-space of the tail words [Liu et al., 2018, Bouraoui et al., 2018]. For instance, in the country-capital relation the head words represent countries while the tail words represent cities. On the other hand, lexicographic relation types do not have specific sub-spaces for the related head and tail words, which means that the offset vectors would not be sufficiently parallel for PairDiff to work well. This is further evident from Figure 2 where the average cosine similarity scores between the relation embeddings computed using PairDiff is significantly smaller for the lexicographic relations compared to that for the encyclopaediac relations on the

BATS dataset. Consequently, the performance of PairDiff on lexicographic relations is poor, whereas MnnPl reports the best results.

As mentioned in Section 2.2, PairDiff is biased towards the attributional similarity between words in two word-pairs compared. To evaluate the effect of this, we group test cases in the DiffVec dataset into two categories: (a) **lexical-overlap** (i.e. there are test cases that have one word in common between two word-pairs) and (b) **lexical-nonoverlap** (i.e. no words are common between the two word-pairs in all the test cases). In other words, given the test word-pair $(a, b)$, then if there is a train word-pair $(a, c)$, $(b, c)$, $(c, a)$ or $(c, b)$ we consider this case in the lexical-overlap set. For example, ($animal$, $cat$) and ($animal$, $dog$) has lexical-overlap because $animal$ is a common word in the two pairs. Figure 3 shows the average 1-NN classification accuracy for the best unsupervised operator PairDiff and **MnnPl**. We see that the performance drops significantly from lexical-overlap to lexical-nonoveralp by ca. 10% for PairDiff, whereas that drop is ca. 1.8% for **MnnPl**. This result indicates that **MnnPl** is affected less by attributional similarity compared to PairDiff.

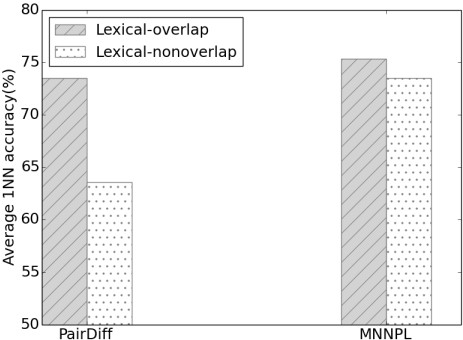

Figure 3: Effect of lexical overlaps in measuring word-pairs relational similarity.

### 4.5 In-domain Performance of the Relation Representation Methods

We evaluate the performance of the relation representation operators considering in-domain setting, wherein we test the performance on relational instances belong to relation types used in the training set. Recall that $\mathcal{R}$ and $\mathcal{D}$ refer to the set of relations and the set of relational instances covering such relations, respectively. In the in-domain setting, we do not need to split $\mathcal{R}$ to source and target relation sets. Instead, we implement 5-stratified folds cross-validation considering the set of relational instances in the dataset $\mathcal{D}$ We use 1-NN and MAP as we did in the out-of-domain experiment. So in-domain experiment setting is very similar to out-of-domain experiment expect in the latter we use $\mathcal{R}_s \neq \mathcal{R}_t$ for the evaluation. Detailed results for in-domain evaluation are presented in Table 4.

### 4.6 Measuring the Degree of Relational Similarity

The relational similarity is the correspondence between the relations of two word-pairs. To measure a relational similarity score between two pairs of words, one must first identify the relation in each pair to perform such comparison. Suitable relation embeddings should highly correlate with human judgments of relational similarity between word-pairs. For this

| | CBOW | | | | SG | | | | GloVe | | | | LSA | | | |
|---|---|---|---|---|---|---|---|---|---|---|---|---|---|---|---|---|
| | DiffVec | | BATS | | DiffVec | | BATS | | DiffVec | | BATS | | DiffVec | | BATS | |
| Method | Acc | MAP | Acc | MAP | Acc | MAP | Acc | MAP | Acc | MAP | Acc | MAP | Acc | MAP | Acc | MAP |
| Super-Concat | 0.698 | 0.51 | 0.600 | 0.481 | 0.649 | 0.463 | 0.449 | 0.342 | 0.695 | 0.513 | 0.485 | 0.374 | 0.680 | 0.501 | 0.413 | 0.358 |
| Super-Diff | 0.583 | 0.43 | 0.424 | 0.329 | 0.538 | 0.397 | 0.322 | 0.238 | 0.564 | 0.404 | 0.349 | 0.263 | 0.560 | 0.382 | 0.325 | 0.25 |
| BiLin | 0.700 | 0.52 | 0.599 | 0.492 | 0.648 | 0.464 | 0.462 | 0.349 | 0.692 | 0.51 | 0.501 | 0.383 | 0.694 | 0.511 | 0.438 | 0.366 |
| PairDiff | 0.686 | 0.386 | 0.484 | 0.329 | 0.621 | 0.334 | 0.399 | 0.263 | 0.662 | 0.371 | 0.442 | 0.288 | 0.642 | 0.339 | 0.398 | 0.279 |
| Concat | 0.717 | 0.385 | 0.417 | 0.284 | 0.673 | 0.336 | 0.344 | 0.24 | 0.672 | 0.349 | 0.385 | 0.261 | 0.667 | 0.345 | 0.344 | 0.26 |
| Add | 0.573 | 0.323 | 0.303 | 0.227 | 0.524 | 0.296 | 0.261 | 0.196 | 0.516 | 0.299 | 0.282 | 0.205 | 0.534 | 0.301 | 0.270 | 0.213 |
| Mult | 0.480 | 0.268 | 0.182 | 0.119 | 0.453 | 0.275 | 0.182 | 0.123 | 0.423 | 0.261 | 0.177 | 0.108 | 0.460 | 0.256 | 0.226 | 0.151 |
| **MnnPl** | **0.797** | **0.656** | **0.60** | **0.483** | **0.765** | **0.619** | **0.497** | **0.379** | **0.796** | **0.655** | **0.531** | **0.416** | **0.785** | **0.639** | **0.479** | **0.387** |

Table 4: 1-NN relation classification results for in-domain setting.

task, we use the dataset proposed by Chen et al. [2017][3] which is inspired by SemEval-2012 task 2 dataset [Jurgens et al., 2012]. In this dataset, humans are asked to score pairs of words directly focusing on a comparison between instances with similar relations. For examples, in Location:Item relation, the pairs (*cupboard*, *dishes*) and (*kitchen*, *food*) are assigned higher relational similarity score (6.18) than the pairs (*cupboard*, *dishes*) and (*water*, *ocean*) which is rated 3.8. Instances of this relation (X, Y) can be expressed by multiple patterns such as "X holds Y" or "Y in the X", and one reason that the second example is assigned low score is that the words in the pair (*water*, *ocean*) are ordered reversely compared to other pairs. Chen et al. [2017] dataset consist of 6,194 word-pairs across 20 semantic relation subtypes. We calculated the relational similarity score of two pairs as the cosine similarity between the corresponding relation vectors generated by the considered operators. Then, we measure the Pearson correlation coefficient between the average human relational similarity ratings and the predicted scores by the methods. For this task, we choose to train the supervised methods on BATS as the overlap of the relation set between BATS and Chen datasets are small. We exclude any word-pairs in Chen dataset that appears in the training data.

Table 5 shows Pearson correlations for all the four embedding models and the relational representation methods across all relations, where high values indicate a better agreement with the human notion of relational similarity. As can be observed, the proposed **Mn-nPl** correlated better with human ratings than the supervised and unsupervised baselines. According to the Fisher transformation test of statistical significant, the reported correlations of **MnnPl** is statistically significant at the 0.05 significant level. Interestingly, the Concat baseline shows a stronger correlation coefficient than PairDiff. Moreover, for SG and LSA embeddings, Add and Mult are considered stronger than PairDiff. In consistent with out-of-domain relation prediction task, CBOW embedding perform better than other embeddings for measuring the degree of relational similarity. Indeed, measuring the degree of relational similarity is a challenging task and required qualified fine-grained relation embeddings to obtain accurate scores of relational instances.

---

3. https://github.com/sdawnchen/vector-space-analogy-analysis

| Method | **MnnPl** | Super-Concat | Super-Diff | BiLin | PairDiff | Concat | Add | Mult |
|--------|-----------|--------------|------------|-------|----------|--------|------|------|
| cbow | **0.309** | 0.228 | 0.163 | 0.258 | 0.172 | 0.277 | 0.223 | 0.204 |
| GloVe | **0.263** | 0.227 | 0.144 | 0.207 | 0.161 | 0.208 | 0.147 | 0.021 |
| SG | **0.251** | 0.213 | 0.113 | 0.176 | 0.161 | 0.208 | 0.147 | 0.021 |
| LSA | **0.266** | 0.171 | 0.098 | 0.199 | 0.154 | 0.245 | 0.197 | 0.190 |

Table 5: Results of measuring relational similarity scores (Pearson's correlations).

## 5. Conclusion

We considered the problem of learning relation embeddings from word embeddings using parametrised operators that can be learnt from relation-labelled word-pairs. We experimentally showed that the penultimate layer of a feed-forward neural network trained for classifying relation types (**MnnPl**) can accurately represent relations between two given words. In particular, some of the disfluencies of the popular PairDiff operator can be avoided by using **MnnPl**, which works consistently well for both lexicographic and encyclopaedic relations. The relation representations learnt by **MnnPl** generalise well to previously unseen (out-of-domain) relations as well, even though the number of training instances is typically small for this purpose.

Our analysis highlighted some important limitations in the evaluation protocol used in prior work for relation composition operators. Our work questions the belief that unsupervised operators such as vector offset can discover rich relational structures in the word embedding space. More importantly we show that simple supervised relational composition operators can accurately recover the relational regularities hidden inside word embedding spaces. We hope our work will inspire the NLP community to explore more sophisticated supervised operators to extract useful information from word embeddings in the future.

Recently, Roller et al. [2018] show that accessing lexical relations such as hypernym relying only on distributional word embeddings that are trained considering 2-ways co-occurrences between words is insufficient. They illustrate the advantages of using the *holistic* (pattern-based) to detect such relations. Indeed, it is expected that the *holistic* and the *compositional* approaches for representing relations have complementary properties since the *holistic* uses lexical contexts in which the two words of interest co-occur, while the *compositional* uses only their embeddings [Shwartz et al., 2016]. Interesting future work includes unifying the two approaches for relation representations.

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
