# OpenReview forum: "Learning Relation Representations from Word Representations"
_AKBC.ws/2019/Conference — AKBC 2019_

### Official Review · AnonReviewer3 · 2019-01-05
**OK work, but the framing makes it sound trivial**

**Rating:** 7
**Confidence:** 3

**Review:**


== Summary ==
The paper proposes a way to embed the relation between a given pair of words (e.g., (aquarium, fish)).
- Assume a dataset of |R| relations (e.g., hypernym, meronym, cause-effect) and many word pairs for each relation.
- In each of the 10 test scenarios, 5 relations are randomly chosen. The test procedure is: for each pair (a, b) with embedding r(a, b), rank (c, d) based on cosine(r(a, b), r(c, d)). Evaluate how well we can retrieve pairs from the same relation as (a, b) (top-1 accuracy and mean average precision).
- The proposed method is to train a multiclass classifier on the |R| - 5 relations. The model is a feed-forward network, and the output of the last hidden layer is used as r(a, b).
- The method is compared with unsupervised baselines (e.g., PairDiff: subtracting pre-trained embeddings of a and b) as well as similar supervised methods trained on a different objective (margin rank loss).

== Pros ==
- The evaluation is done on relations that are not in the training data. It is not trivial that a particular relation embedding would generalize to unseen and possibly unrelated relations. The proposed method generalizes relatively well. Compare this to the proposed margin rank loss objective, which performs well on the classification task (Table 7) but is worse than PairDiff on test data.

== Cons ==
- The paper is framed in such a way that the method is trivial. My initial thought from the abstract was: "Of course, supervised training is better than unsupervised ones", and the introduction does not help either. The fact that the method generalizes to unseen test relations, while a different supervised method does to a lesser extent, should be emphasized earlier.
- The reason why the embedding method generalizes well might have something to do with the loss function used rather than how the word vectors are combined (difference, concatenation, bilinear, etc.). This could be investigated more. Maybe one loss is better at controlling the sizes of the vectors, which is an issue discussed in Section 2.2.
- The result on the DiffVec dataset is pretty low, considering that a random baseline would get an accuracy of ~20% (1 in 5 classes). The results also seem to be only a bit better than the unsupervised baseline PairDiff on the BATS dataset.
- The writing is a bit confusing at times. For instance, on page 12, for the "lexical-overlap" category, it should not be possible for any two test pairs to have exactly 1-word overlap, or maybe I am missing something.

---

> ### Author Response · Authors · 2019-01-30
> **Response to Reviewer 3**
>
> We appreciate your effort in reviewing our work.
> == Cons ==
> 1-- Thanks for your comment. We will add couple of sentences in the abstract and the introduction to emphasize the generalization ability of the proposed model.
>
> 2-- The supervised baselines with different inputs (PairDiff, Concat and BiLin) that are described in Section 4.4 are all trained using the same loss: ranking loss defined in equation 5.
> -	We perform a direct comparison between two ways to derive relation representations: the penultimate layer of a neural network trained on relation prediction task using the softmax cross-entropy loss among relations vs direct relation representation learning models using analogous and non-analogous pairs of word-pairs considering the ranking loss. In this case, the two models are exactly same including the number of hidden layers, non-linear activation etc, with only a change in the loss. Empirically the proposed MnnPL (softmax cross entropy) outperforms the direct relation representation trained on the pairs of word-pairs relational data.
> -	In fact, there are many objective functions that can be used to train a supervised model for relation representations which we can evaluate them extensively in our future work.
>
> 3-- In Table 2, MnnPL outperforms the unsupervised baselines for BATS but the gap is smaller than those in DiffVec. In BATS, we got two semantic relation types as shown in Table 3. Breaking down the performance of encyclopaedic and lexicographic indicates that MnnPL performs significantly better the PairDiff for lexicographic relations and slightly better for encyclopaedic ones. Discussion about the two relation types with some justification are illustrated in Page 12 (Figure 2).
> -	As shown in Appendix A (table 5), all the semantic relations in DiffVec are lexicographic type of relations. Even though the comparison made for BATS dataset show the difficulty of capturing lexicographic relations compared to encyclopaedic one, it is shown that MnnPL reports 48.6% accuracy with CBOW embeddings (statistically significant that the random baseline).
>
> 4-- If the test word-pair is (a,b), then if there is a train word-pair (a,c), (b,c), (c,a) or (c,b) then we say there is a lexical-overlap between the test and the train datasets. For example, (‘animal, cat) and (animal, dog) has lexical-overlap because animal is a common word in the two pairs. We will add this clarification for lexical-overlap in the revised version.

---

### Official Review · AnonReviewer1 · 2019-01-09
**Elegant Method, Neat Result, Manuscript Needs Major Reorganization**

**Rating:** 7
**Confidence:** 4

**Review:**

This paper presents a simple model for representing lexical relations as vectors given pre-trained word embeddings.

Although the paper only evaluates on out-of-context lexical benchmarks, as do several other papers in lexical-semantics, the empirical results are very encouraging. The proposed method achieves a substantial gain over existing supervised and unsupervised methods of the same family, i.e. that rely only on pre-trained word embeddings as inputs.

In my view, the main innovation behind this method is in the novel loss function. Rather than using just a single word pair and training it to predict the relation label, the authors propose using *pairs of word pairs* as the instance, and predicting whether both pairs are of the same relation or not. This creates a quadratic amount of examples, and also decouples the model from any schema of pre-defined relations; the model is basically forced to learn a general notion of similarity between relation vectors. I think it is a shame that the loss function is described as a bit of an afterthought in Section 4.4. I urge the authors to lead with a clear and well-motivated description of the loss function in Section 3, and highlight it as the main modeling contribution.

I think it would greatly strengthen the paper to go beyond the lexical benchmarks and show whether the learned relation vectors can help in downstream tasks, such as QA/NLI, as done in pair2vec (https://arxiv.org/abs/1810.08854). This comment is true for every paper in lexical semantics, not only this one in particular.

It would also be nice to have an empirical comparison to pattern-based methods, e.g. Vered Shwartz's line of work, the recent papers by Washio and Kato from NAACL 2018 and EMNLP 2018, or the recently-proposed pair2vec by Joshi et al (although this last one was probably published at the same time that this paper was submitted). The proposed method doesn't need to be necessarily better than pattern-based methods, as long as the fundamental differences between the methods are clearly explained. I think it would still be a really exciting result to show that you can get close to pattern-based performance without pattern information.

My main concern with the current form of the paper is that it is written in an extremely convoluted and verbose manner, whereas the underlying idea is actually really simple and elegant. For example, in Section 3, there's really no reason to use so many words to describe something as standard as an MLP. I think that if the authors try to rewrite the paper with the equivalent space as an ACL short (around 6-7 pages in AKBC format), it would make the paper much more readable and to-the-point. As mentioned earlier, I strongly advise placing more emphasis on the new loss function and presenting it as the core contribution.

Minor comment: Section 2.2 describes in great detail the limitation of unsupervised approaches for analogies. While this explanation is good, it does not properly credit "Linguistic Regularities in Sparse and Explicit Word Representations" (Levy and Goldberg, 2014) for identifying the connection between vector differences and similarity differences. For example, the term 3CosAdd was actually coined in that paper, and not in the original (Mikolov, Yih, and Zweig; 2013) paper, in order to explain the connection between adding/subtracting vectors and adding/subtracting cosine similarities. The interpretation of PairDiff as a function of word similarities (as presented in 2.2) is very natural given Levy and Goldberg's observation.

---

> ### Author Response · Authors · 2019-01-30
> **Response**
>
> We would like to thank the reviewer for the useful comments. We will take your suggesstions about organizing the paper in the final version.

---

### Official Review · AnonReviewer2 · 2019-01-10
**Novel solution and comprehensive experimentation**

**Rating:** 7
**Confidence:** 4

**Review:**

This paper proposes a novel solution to the relation composition problem when you already have pre trained word/entity embeddings and are interested only in learning to compose the relation representations . The proposed supervised relational composition method learns a neural network to classify the relation between any given pair of words and as a by-product, the penultimate layer of this neural network, as observed by the authors, can be directly used for relation representation.
The experiments have been performed on the BATS dataset and the DiffVec Dataset.
The inferences that were made to advocate for the usefulness of proposed MnnPL are as follows-
- Out_of_domain relation prediction experiment to test for generalisability showed that MnnPL outperformed other baselines at this task
- The interesting analysis in table 3 highlights the difficulty in representing lexicographic relations as compared to encyclopedic. MnnPL outperforms others here too. The authors provide a reasonable explanation (Figure 2) as to why the PairDiff operator that was proposed to work well on the Google analogy dataset works worse in this scenario.
- The experiment to measure the degree of relational similarity using Pearson correlation coefficient, showcases that the relational embeddings from MnnPL are better correlated with human notion of relational similarity between word pairs
- They also showed that MnnPl is less biased to attributional similarity between words
The authors show that the proposed MnnPL had outperformed other baselines on several experiments.

Some of the positive aspects about the paper
- elaborately highlighted all the implementations details in a crisp manner
- Extensive experimentation done and a very due-diligent evaluation protocol.
- In the experiments the authors have compared their proposed supervised operator extensively with other unsupervised operators like PairDiff,  CONCAT, e.t.c and some supervised operators like SUPER_CONCAT, SUPER_DIFF e.t.c. They also compared against the bilinear operator proposed by Hakami et al., 2018, which was published very recently.
Some of the limitations of the work
- Though extensive experiments have been done and elaborate evaluation  protocols have been followed to advocate for MnnPL, I believe that it lacked slightly on novelty side.
- Reference to Table 2 on page 12 should actually be a reference to figure 2

Questions for rebuttal:-
- Some reasoning on why does the CONCAT baseline show a better Pearson correlation coefficient than PairDIff?
- Interesting to see that CBOW performed better than others, especially GLOVE, on all experiments. Some analysis on this.
- Break down of the performance for the two semantic relation types, could be shown on a few other datasets to strengthen claim.

---

> ### Author Response · Authors · 2019-01-30
> **Response to Reviewer 1**
>
> We want to thank the reviewer for the time and review.
> Questions for rebuttal:
> 1-- CONCAT is a general operator that maps a pair to a higher dimensional space that considers the features of the two words in the word-pairs. We empirically observe that CONCAT operator for relations is significantly affected by the pairwise similarities of the corresponding words a and c, b and d for (a, b) and (c, d) word-pairs [1]. For relation prediction task in our work, we alleviate the effect of getting high relational similarity scores due to attributional similarities by excluding the 25% nearest pairs for each stem using cross-pair attributional similarity scores given by Equation 6. However, in measuring the degree of relational similarities and compare them with the human scores we did not consider such pre-processing step because of the nature of this evaluation is different than the relation prediction. In Chen dataset, there are many word-pairs that are assigned higher relational scores (human scores) than others because of the similarities between the corresponding entities in pairs. For example, (animal, pig) and (insect, ant) got 6.33 compared to (animal, pig) and (song, opera) that scored 4.3. It was also consistent with a closely related task measuring degrees of prototypicality in a recent work by Jameel, et al (2018) wherein PairDiff and Concat report 0.173 and 0.167 Spearman correlations, respectively [2].
>
> 2--	First we want to note that this result is consistent with a recent comparative study by Hakami and Bollegala (2017). In [3] it has been shown that CBOW embeddings with unsupervised PairDiff operator perform the best for relation representations compared to various embedding types among many relational datasets. Similarly, Levy et al. (2015) show that CBOW (and SG) are better than GloVe in most cases for Google and MSR relational dataset using 3CosAdd and 3CosMult pair-based operators for answering analogical questions [4].
> -	The GloVe objective for learning word embeddings aims to reduce the difference between the dot product of the vectors of the two words and the log of their co-occurrence. Thus, GloVe model is trained on the word-context co-occurrence matrix instead of the text corpus. According to this, GloVe is close to traditional word representation methods such as LSA that factorizing a word-context co-occurrence matrix. On the other hand, the CBOW objective seeks to predict a target word from its surrounding context. This difference in the objective between GloVe and CBOW might make sense for the superiority of CBOW prediction word embedding to capture informative information to induce relations between words.
> -	This work was not aimed to compare the embedding types, instead, we wanted to show that the proposed MnnPL method for representing relations is performing well across various type of word embeddings. Despite this, we train all the models on the same corpus to make a fair comparison to some extent. It is also known that hyperparameters tuning and pre-processing (or post-processing) are affecting the performance of such word embedding models. We also carry out our experiments using the 300 dimensional pre-trained publicly available GloVe and CBOW embeddings and it turns out that again CBOW outperforms GloVe for our tasks.
>
> 3--	In BATS dataset, the semantic relations are classified to encyclopaedic and lexicographic where we show the performance on each relation type. On the other hand, as shown in Appendix A (table 5), all the semantic relations in DiffVec are lexicographic.
>
> References:
> [1] Turney, Peter D. "Domain and function: A dual-space model of semantic relations and compositions." Journal of Artificial Intelligence Research 44 (2012): 533-585.
> [2] Jameel, Shoaib, Zied Bouraoui, and Steven Schockaert. "Unsupervised Learning of Distributional Relation Vectors." Proceedings of ACL, Melbourne, Australia (2018).
> [3] Hakami, Huda, and Danushka Bollegala. "Compositional approaches for representing relations between words: A comparative study." Knowledge-Based Systems 136 (2017): 172-182.
> [4] Levy, Omer, Yoav Goldberg, and Ido Dagan. "Improving distributional similarity with lessons learned from word embeddings." Transactions of the Association for Computational Linguistics 3 (2015): 211-225.

---

### Meta-Review · Area_Chair1 · 2019-02-11
**Interesting method and results, but presentation could be improved**

**Recommendation:** Accept (Poster)
**Confidence:** 4

**Metareview:**

This paper presents an approach for learning relation embeddings that are a feed forward function of the embeddings of a word pair. The method uses a novel training objective and is shown to work well on out-of-context lexical benchmarks. The paper could be framed better and there are a number of points where the reviewers needed clarifications, but the authors have promised revisions to address these concerns.

---

### Decision · Program_Chairs · 2019-02-15
**AKBC 2019 Conference Decision**

Accept